# Progressive Immunodeficiency with Gradual Depletion of B and CD4^+^ T Cells in Immunodeficiency, Centromeric Instability and Facial Anomalies Syndrome 2 (ICF2)

**DOI:** 10.3390/diseases7020034

**Published:** 2019-04-04

**Authors:** Georgios Sogkas, Natalia Dubrowinskaja, Anke K. Bergmann, Jana Lentes, Tim Ripperger, Mykola Fedchenko, Diana Ernst, Alexandra Jablonka, Robert Geffers, Ulrich Baumann, Reinhold E. Schmidt, Faranaz Atschekzei

**Affiliations:** 1Department of Clinical Immunology and Rheumatology, Hannover Medical School, 30625 Hannover, Germany; Dubrowinskaja.Natalia@mh-hannover.de (N.D.); Ernst.Diana@mh-hannover.de (D.E.); Jablonka.Alexandra@mh-hannover.de (A.J.); Schmidt.Reinhold.Ernst@mh-hannover.de (R.E.S.); Atschekzei.Faranaz@mh-hannover.de (F.A.); 2Department of Human Genetics, Hannover Medical School, 30625 Hannover, Germany; Bergmann.Anke@mh-hannover.de (A.K.B.); Lentes.Jana@mh-hannover.de (J.L.); Ripperger.Tim@mh-hannover.de (T.R.); 3Institute of Pathology, Hannover Medical School, 30625 Hannover, Germany; Fedchenko.Mykola@mh-hannover.de; 4Helmholtz Centre for Infection Research, 38124 Braunschweig, Germany; Robert.Geffers@helmholtz-hzi.de; 5Department of Paediatric Pulmonology, Allergy and Neonatology, Hannover Medical School, 30625 Hannover, Germany; Baumann.Ulrich@mh-hannover.de

**Keywords:** ICF syndrome, B cell immunodeficiency, T cell immunodeficiency, combined immunodeficiency, ICF2, ZBTB24

## Abstract

Immunodeficiency, centromeric instability and facial anomalies syndrome 2 (ICF2) is a rare autosomal recessive primary immunodeficiency disorder. So far, 27 patients have been reported. Here, we present three siblings with ICF2 due to a homozygous *ZBTB24* gene mutation (c.1222 T>G, p. (Cys408Gly)). Immune deficiency in these patients ranged from late-onset combined immunodeficiency (CID) with severe respiratory tract infections and recurrent shingles to asymptomatic selective antibody deficiency. Evident clinical heterogeneity manifested despite a common genetic background, suggesting the pathogenic relevance of epigenetic modification. Immunological follow-up reveals a previously unidentified gradual depletion of B and CD4^+^ T cells in all three presented patients with transition of a common variable immunodeficiency (CVID)-like disease to late-onset-CID in one of them. Considering all previously published cases with ICF2, we identify inadequate antibody responses to vaccines and reduction in CD27^+^ memory B cells as prevalent immunological traits. High mortality among ICF2 patients (20%) together with the progressive course of immunodeficiency suggest that hematopoietic stem cell transplantation (HSCT) should be considered as a treatment option in due time.

## 1. Introduction

The immunodeficiency-centromeric instability-facial anomalies (ICF) syndromes are characterized by humoral or combined immunodeficiency, facial dysmorphism, variable intellectual deficit and chromosomal abnormalities [1,2,3]. The latter involve CpG hypomethylation in the heterochromatic regions of chromosomes 1, 9 and 16 [4,5]. So far 4 types of ICF syndromes have been identified [3]. ICF1 is caused by biallelic mutations in *DNMT3B*, which encodes the DNA methyltransferase 3B [6] and is the most frequent type of ICF syndrome, accounting for approximately half of patients. ICF2 accounts for approximately 30% of ICF patients and is caused by biallelic mutations in *ZBTB24*, which encodes the zinc-finger-and BTB-domain containing 24 protein [3,7]. More recently, two new types of ICF syndromes, ICF3 and ICF4, have been reported, due to mutation in *CDCA7* and *HELLS*, respectively [8]. With the exception of reports on a milder intellectual impairment in ICF1 and a gender bias in ICF2, with most patients being males, there are no significant clinical differences between these two main subtypes of ICF syndromes [3,9]. Immunological findings appear to be similar in ICF1 and ICF2, though humoral immunodeficiency has been reported to be more severe in ICF1 [3].

The mechanism by which ICF underlying mutations result in loss of DNA methylation remains unknown. A recent work on comparative methylome analysis of patients with different ICF subtypes identified heterochromatic loci, whose methylated state specifically relies on either ZBTB24 or DNMT3B [10]. ZBTB24 belongs to a large family of transcription factors, including more than 40 different proteins, which form homo- or heterodimers in the nucleus via their protein-interacting BTB domain and bind to target genes via their DNA-binding C_2_H_2_ zinc-finger domains [10,11,12]. ZBTB24 is highly expressed by B-cells. Silencing its expression in a human B-cell line resulted in reduced proliferation by blocking cell cycle progression from the G0/1- to S-phase [11]. In these cells, cell cycle arrest may have been the consequence of enhanced expression of IRF-4, which negatively regulates B cell proliferation. These findings may provide an explanation for humoral immunodeficiency in ICF2. Further, loss of DNA methylation as a consequence of disease-causing *ZBTB24* variants has been identified among else in genes implicated in neuronal development, which might explain intellectual abnormalities in ICF2 [10].

Immunodeficiency in ICF2 syndrome manifests with recurrent respiratory and/or gastrointestinal infections as a consequence of hypo- or agammaglobulinemia [3]. However, reports on fatal opportunistic fungal and/or viral infections as well as immunological profiles with T cell defects in some patients with ICF2 suggest that ICF2 can manifest as a combined immunodeficiency [3,13].

Here, we present three siblings with ICF2, including two dizygotic twin sisters. Despite their common genetic background, including the same causative mutation in *ZBTB24*, clinical heterogeneity and evident differences in immunological profiles among these patients suggest the pathogenic relevance of epigenetic modification in this monogenic immunodeficiency disorder. We further provide a long follow-up including repeated immunological investigations, which reveals a progressive course of immunodeficiency in all three patients, suggesting the need for close patient monitoring and early consideration of hematopoietic stem cell transplantation (HSCT).

## 2. Materials and Methods

### 2.1. Genetic Analysis

Blood samples were collected in the outpatient clinic for Immunology/Rheumatology of Hannover University Hospital. Genomic DNA (gDNA) was isolated from peripheral whole blood using QIAamp DNA Blood Midi Kit, according to the manufacture’s protocol (Qiagen, Venlo, The Netherlands). Whole exome sequencing (WES) was performed on genomic DNA samples from patients P2, P3 and their father at the Helmholtz Centre for Infection Research (HZI), Germany. Concentration and quality of the purified genomic DNA (gDNA) was determined with an Agilent Technologies 2100 Bioanalyzer (Agilent Technologies, Santa Clara, CA, USA). The DNA sequencing library consisted of 100 ng fragmented gDNA and was generated with Agilent SureSelectXT Reagent Kits v5 UTR (70 Mb) according to the manufacturer’s protocols (Illumina, San Diego, CA, USA). Libraries were sequenced on an Illumina HiSeq2500 platform using TruSeq SBS Kit v3-HS (200 cycles, paired end run) with an average of 12.5 × 10^6^ reads per single exome (mean coverage: 50X). The GATK-Pipeline (GenomeAnalysisTK-1.7) was applied for read quality trimming, read alignment to reference (GRCh37/hg19) and quality trimmed variant calling. Variant annotation was performed using ANNOVAR. We selected for rare variants with low minor allele frequency (MAF < 0.05). Sanger sequencing was performed as described elsewhere [14] to validate the identified rare *ZBTB24* variant and its co-segregation with disease phenotype in this family using the primers: F 5′-GAGGAGGCAGTGAGTTGAGG-3′ and R 5′-GGGACAGACGAGATGGAGTT-3. For karyotype analysis phytohaemagglutinin (PAH)-stimulated whole blood cell cultures were used. Chromosome preparations and fluorescence R-banding was performed as describes elsewhere [15].

This study was conducted in accordance with the Declaration of Helsinki and was also approved from the Ethic committee of the Hannover Medical School (approval number: 2189). All patients and their parents signed an informed consent form. 

### 2.2. Phenotypic Analyses of Lymphocytes

Peripheral blood mononuclear cells (PBMCs) were obtained from heparinized peripheral blood samples of healthy consenting donors and patients by centrifugation over a ficoll hypaque gradient. Phenotypic analyses were performed as multicolour immunofluorescence of Ficoll–Hypaque-separated cell samples utilizing directly labeled monoclonal antibodies as described previously [16]. Briefly, 1 × 10^5^ to 2 × 10^6^ cells/well were incubated with murine monoclonal antibodies against the appropriate antigens at an optimal dilution for 20 min at 4 °C. Nonspecific binding was eliminated by mixing the samples with a 1:5 (*v*/*v*) solution of a commercial human IgG (Octagam; Octapharma, Langenfeld, Germany). Samples were washed three times in PBS/BSA, and at least 10^4^ cells per appropriate gate were analyzed using a FACSCanto II flow cytometer with Cell Quest software (Becton Dickinson, Heidelberg, Germany). Offline data analysis was performed by using FCS Express software V6 (Denovo Software, Glendale, CA, USA). The following antibodies (all obtained from Biolegend, London, UK, if not otherwise stated) were used for this study: CD3 PerCP (BD Pharmingen, Heidelberg, Germany), CD3 PE, CD3 APC, CD4 FITC (Beckman Coulter, Krefeld, Germany), CD4 PerCP, CD8 PE, CD8 PECy7, CD16 FITC, CD19 BV510, CD21 PE, CD24 FITC, CD27 FITC, CD28 BV421, CD31 FITC, CD38 PECy7, CD45R0 BV421, CD45RA BV510, CD56 APC, CD56 BV421, CD335 PE, CCR5 Alexa Flour 647, CCR7 Alexa Flour 647, Granzyme A FITC (BD Pharmingen), Granzyme B Pacific Blue, Granzyme K PE (Immunotools, Friesoythe, Germany), Perforin BV510, IFNγ Alexa Flour 647 (BD Pharmingen), IgD PE, IgM Alexa Flour 647, IL-17A PE (BD Pharmingen). Each flow cytometric analysis was controlled with appropriate isotype-matched mAbs.

### 2.3. Proliferation Assays

For standard proliferation assays 2 × 10^5^ PBMCs were cultured in R10 medium (composed of RPMI 1640 media (Biochrom, Berlin, Germany) supplemented with 10% FCS, 1 mM l-glutamine, 50 U/mL penicillin, 0.5 mM sodium pyruvate, and 50 μg/mL streptomycin) for 48 h with PHA, ConA, PWM, and immobilized CD3 mAb (iCD3) either with or without soluble co-stimulatory anti-CD28 (1 μg/mL), respectively. For immobilization 100 µL anti-CD3 (OKT3, 2.5 μg/mL; purified from hybridoma supernatant) was adsorbed onto Nunc Maxi-Sorb 96-well plates (Sigma Aldrich, St. Louis, MO, USA) for 60 min at 37 °C. The plate was washed three times. Proliferation of CD4^+^ T cells was assessed after magnetic enrichment of the target cells from frozen PBMC after thawing using the IMag CD4 T Lymphocyte Enrichment Set (Becton Dickinson, Heidelberg, Germany) and incubating 5 × 10^4^ cells/well with media, PHA and iCD3 either with or without soluble co-stimulatory anti-CD28 (1μg/mL), respectively, in triplicates for 72 h. Radioactive thymidine was added to each well. 24 h later cells were harvested and incorporated radioactivity was determined using a beta-counter.

### 2.4. Cytotoxicity Assay

Cytotoxicity of NK cells was determined by 4 h chromium (^51^Cr) release assay against K562 target cells following the protocol as described elsewhere [17]. Briefly, K562 cells were labelled with radioactive ^51^Cr by incubating the cells with 3 MBq Na^51^CrO_4_ for 1 h at 37 °C. The cells were washed twice and plated in triplicates in V-bottom 96-well plates. Effector: Target ratios of 20:1, 10:1, 5:1 and 2.5:1 were used. NK cell mediated lysis of targets results in the release of ^51^Cr into the supernatant. Following the 4 h incubation, the plates were centrifuged. A total of 25 μL cell-free supernatant was collected from each well, and the ^51^Cr release was measured by a gamma counter.

## 3. Results

### 3.1. Clinical Findings

All three patients are siblings born to healthy non-consanguineous German parents (Figure 1).

P1 is a 28 year old male. He was born small for gestational age (i.e., 1800 g, 39th week of pregnancy, <1st percentile) and had a severe sepsis during the first days of life. After successful antibiotic treatment he developed a chronic cholestatic hepatitis, whose etiology remained unclear despite laboratory investigations and a liver biopsy, which was performed at the age of 13 months. Treatment with cholestyramine led to a substantial decrease in liver enzymes, which since then remained close to normal values despite small fluctuations and did not increase even after suspending cholestyramine treatment, two years later. Since the first year of life he presented recurrent skin rash diagnosed as atopic dermatitis. At the age of 2 years, he suffered from a severe pneumonia and thereafter he had recurrent respiratory tract infections including two additional pneumonias until the age of 4 years (one due to *H. influenza*). Immunological investigations at that time revealed profound hypogammaglobulinemia with IgG and IgM deficiency and absence of antibody responses to vaccination against tetanus, diphtheria and *H.influenzae* type B. Immunophenotyping of blood lymphocyte subsets revealed normal B and T cell counts. These findings were consistent with the diagnosis of common variable immunodeficiency (CVID). Immunoglobulin replacement therapy, which was introduced at that time, resulted in a substantial reduction in infection frequency, preventing further pneumonias. The patient developed at the age of 25 years a persistent CD4^+^ T cell lymphocytopenia (CD4^+^ T cells < 200/µL) and has been since then prophylactically treated with co-trimoxazole and fluconazole.

P2 is the four years younger sister of patient P1. She and her twin sister were born in the 34th week of gestation with normal weight for this gestational age. At the age of 3 years, P2 was diagnosed with selective mutism. Since the age of four years she suffered from recurrent upper respiratory tract infections. At the age of 5 years she had severe chickenpox. At the age of 7 years she presented measles-like rash and prolonged fever after vaccination with MMR and later at same age she was diagnosed with pneumonia in the right middle lobe. Consolidation persisted in chest radiograph despite antibiotic regimens, so that a bronchoscopy was performed. Atypical mycobacteria were identified in the bronchoalveolar lavage (BAL) by polymerase chain reaction (PCR). Identification of mycobacterial strain was not available at that time. Acid-fast stain and culture for mycobacteria remained negative. The atypical mycobacterial infection was treated with rifampicin, ethambutol and azithromycin for nine months. Despite an initial regression, the pulmonary infiltrate persisted, so that a middle lobe resection was performed. Histological examination revealed a focal bronchial malformation with bronchiole-like structures and microcysts lined by ciliary respiratory epithelium, which however lacked cartilage and bronchial glands. Thereafter the patient experienced no further severe or recurrent infections with the exception of four episodes with shingles, which were localized to one or two adjacent dermatomes and were treated with either acyclovir or valaciclovir.

P3, the twin sister of P2, was also diagnosed with selective mutism in early childhood. In contrast to her siblings, she presented no significant or recurrent infections. Since the age of 7 years we have documented a single severe bronchitis and a pharyngitis, which were treated with conventional antibiotics. At the age of 24, she suffered a focal onset secondarily generalized seizure and is thereafter treated with lamotrigine. Cerebrospinal fluid examination and neuroimaging (CT and MRI scan of brain) revealed no abnormalities.

Table 1 summarizes all characteristics of patients.

### 3.2. Immunological Investigations

Immunological investigations revealed IgM deficiency with normal IgA levels in all three patients (Table 2). Further, all three patients displayed an absence of specific antibodies against recall antigens. Only one of them, P1, had a clinically relevant hypogammaglobulinemia with severe IgG deficiency. P1 presented a near absence of circulating B lymphocytes, whereas P2 and P3 had normal, though declining absolute counts of CD19^+^ B cells (Figure 2). Further analysis of peripheral blood B cell subsets revealed a substantial reduction in CD27^+^ memory B cells in all three patients. Regarding T cells, all three patients displayed a progressive reduction in CD4^+^ T cell counts, which was extremely pronounced in case of the male patient (Figure 2). At first diagnosis of primary immunodeficiency the male patient presented normal T cell counts and a subsequently performed peripheral blood lymphocyte phenotyping showed normal CD4^+^ T cell and CD45RA^+^ naïve CD4^+^ T cell counts. However, subsequent measurements of peripheral blood T cell subsets, including a recent peripheral lymphocyte phenotyping revealed a gradual reduction in both CD4^+^ T cell and CD45RA^+^ naïve CD4^+^ T cell counts, which is consistent with a transition to a combined immunodeficiency disorder. T cell function was assessed in vitro in all three patients revealing normal T cell proliferation to mitogens and tuberculin purified protein derivative (PPD). With respect to the NK cells, all three patients presented relatively low NK cells counts and a considerably reduced natural and antibody-dependent cytotoxicity. Immunological investigations of the three studied ICF2 patients are summarized in Table 2 and Appendix A.

### 3.3. Identification of A Pathogenic Mutation in ZBTB24

Whole-exome sequencing was performed in both the two dizygotic twin sisters (P2 and P3) and their father. A missense variant in *ZBTB24* was identified. In particular, we identified a homozygous previously described deleterious missense variant, NM_014797: c.1222 T>G (p.Cys408Gly), in *ZBTB24* in both twin sisters [7,18]. Their father was found to be a heterozygous carrier. Sanger sequencing identified the same homozygous mutation in P1 and revealed that patients’ mother is a heterozygous carrier as well. The T>G substitution results in an amino acid change in the zinc finger domain affecting a highly conserved cysteine in a C2H2-type 5 zinc finger motif of ZBTB24 [18]. Evident phenotypic heterogeneity among the here described and previously reported patients with the c.1222 T>G variant suggests the absence of a univocal genotype-phenotype correlation.

## 4. Discussion

We present here three new cases with ICF2, a rare primary immunodeficiency syndrome due to a homozygous mutation in *ZBTB24* (i.e., NM_014797: c.1222 T>G (p.Cys408Gly)). The combination of all three, facial abnormalities, compromised humoral immunodeficiency and centromeric instability, forms the diagnostic triad of ICF [9] and should urge physicians to genetic testing, focused on the four so far identified disease genes (*DNMT3B*, *ZBTB24*, *CDCA7* and *HELLS*). Lack of evident humoral immunodeficiency in two of the here presented patients and our not being familiar with the facial abnormalities of ICF led to diagnosis though the more elaborate genetic approach of WES. The identified mutation has been previously reported in three patients with ICF2, one of whom—similar to the here presented patients—was homozygous and the other two were compound heterozygous [7,18]. German origin of all ICF2 patients with this mutation suggests a founder effect. Variable manifestation of ICF2 among our patients, despite the same deleterious mutation and common genetic background, reveals the lack of an univocal genotype-phenotype correlation and suggests the pathogenic role of epigenetic modification in ICF2.

Humoral immunodeficiency with pronounced hypogammaglobulinemia appears common among patients with ICF2 [3,7,13]. However, only one among the here presented patients had clinically relevant hypogammaglobulinemia with severe IgG deficiency. To our knowledge, so far 27 patients with ICF2 have been reported (Appendix A). A review of existing data on all known ICF2 patients, including the here presented ones, reveals that most of them (23/30, i.e., 77%) presented IgG deficiency and nearly all presented IgM deficiency (28/30, i.e., 93%) (Figure 3). In vivo production of antibodies has been evaluated only in case of three previously reported patients [18,19,20] who all—similar to our patients—presented no or inadequate responses to vaccinations. Low B cell counts have been so far reported for 11 out of 27 patients for whom data are available (i.e., 41%) [9,13,18,19,20,21,22,23] including the here reported patient with hypogammaglobulinemia. Another rarely evaluated, though common abnormality among previously reported ICF2 patients, which we confirmed in our patients, is the substantially reduced CD27^+^ memory B cell count. These cells have so far been evaluated in 12 patients [3,13,23], all of whom presented reduced counts (Figure 3).

A detailed evaluation of cellular immunity is missing in case of most of the 27 previously reported patients. For most of them for whom data are available, T cell counts have been reported to be normal. Enhanced percentages of CD8^+^ T cells appear to be relatively common and are reported for eight among 23 patients for whom data are available (35%) [9,13,18,19,20,21,22]. Increased percentages of CD8^+^ T cell counts appear in most cases to be the consequence of increased absolute counts of CD8^+^ T cells in the absence of a relevant CD4^+^ T cell reduction. Among the here presented cases, one patient presented a significant CD4^+^ T cell lymphopenia, indicating a late-onset CID. To our knowledge this is the first ICF2 patient reported to have a significant CD4^+^ T cell lymphopenia, falling under CID (<200/µL). With respect to the NK cells, all patients presented here had low counts with reduced natural and antibody-dependent cytotoxicity. Responses of T cells to mitogens or antigens are reported for 11 patients and defects have been reported in 6 of them [3,13,18,20]. Here we detected adequate T cell responses to mitogens in all three reported patients. NK cell counts or function had been so far evaluated in 20 patients and only eight of them [3,13,18,19,21,22] including the here presented patients, displayed defects (i.e., 40%).

More importantly, follow-up immunological investigations reveal that, despite previously highlighted phenotypic and immunological discrepancies among the here presented patients, all three of them display a progressive immunodeficiency with gradual depletion of B lymphocytes and CD4^+^ T cell counts (Figure 2). This stresses the necessity of regular follow-up immunological investigations in patients with ICF2, as early diagnosis of CID may prevent fatal opportunistic infections.

Facial abnormalities have been reported in all 30 known patients with ICF2. A description is available for 26 patients (Appendix A). Broad nasal bridge and hypertelorism appear to be the most common facial features of ICF2 (20/26, i.e., 77% and 17/26, i.e., 65%, respectively). Other non-facial dysgenetic traits of ICF2 are rare and include hypospadia, cryptorchidism and brain anomalies [3,24,25]. Here we describe a novel dysgenetic trait, bronchial malformation with absence of bronchial glands and cartilage, in one of the three presented patients. Overall, despite the preeminence of some facial features, there is no recognizable phenotypic presentation of ICF2 patients.

The most commonly reported infectious manifestations of ICF2 are respiratory tract infections as a consequence of antibody failure (Figure 3). However, opportunistic infections like candidiasis, *Pneumocystis* pneumonia and progressive multifocal leukoencephalopathy (PML) have been reported [7,13]. EBV or CMV viremia and infections appear to be relatively common [18,20,22]. Overall, despite the short documented follow-up period, ICF2 appears to have a high mortality rate of 20 % (6 patients with deathly outcome out of all known 30 patients), mostly stemming from infectious complications (documented for 5 out of 6 patients, Appendix A). Considering this high mortality rate, previous reports on severe complications in patients with ICF2, such as lymphomas and EBV-induced hemophagocytic lymphohistiocytosis [7,9] as well as the here presented evidence on a progressive impairment of the immune system, we suggest early consideration of HSCT in all ICF2 patients.

## Figures and Tables

**Figure 1 diseases-07-00034-f001:**
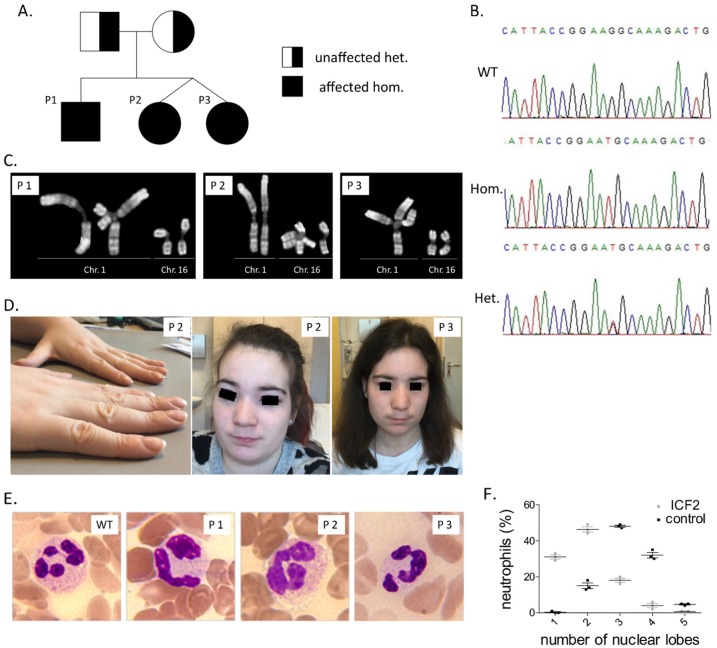
Segregation of *ZBTB24* c.1222 T>G mutation in studied patients’ family. Pedigree (**A**) and representative sequencing chromatograms (**B**) are depicted. (**C**) Representative images of all three studied patients showing spontaneous under-condensation of subcentromeric constitutive heterochromatin of chromosomes 1 and 16 as well as multibranched chromosomes. (**D**) Flat nasal bridge in patients P2 and P3 and nail clubbing in patient P3. (**E**) Blood smears stained with hematoxylin-eosin, depicting pseudo-Pelger–Huët anomaly of neutrophils, magnification: 100×. (**F**) Dot plot of nuclear lobe numbers of neutrophils of the three patients and three healthy blood donors.

**Figure 2 diseases-07-00034-f002:**
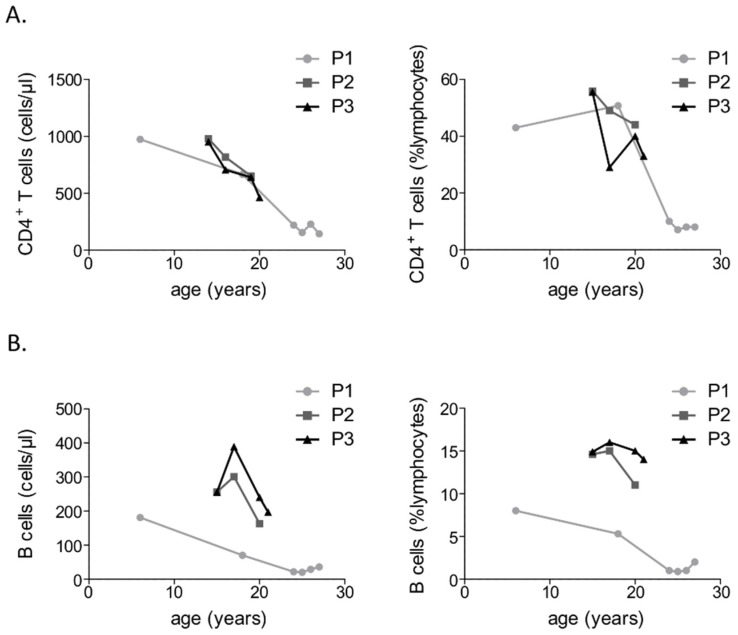
Progressive depletion of CD4^+^ T cells (**A**) and CD19^+^ B cells (**B**) in three patients with ICF2.

**Figure 3 diseases-07-00034-f003:**
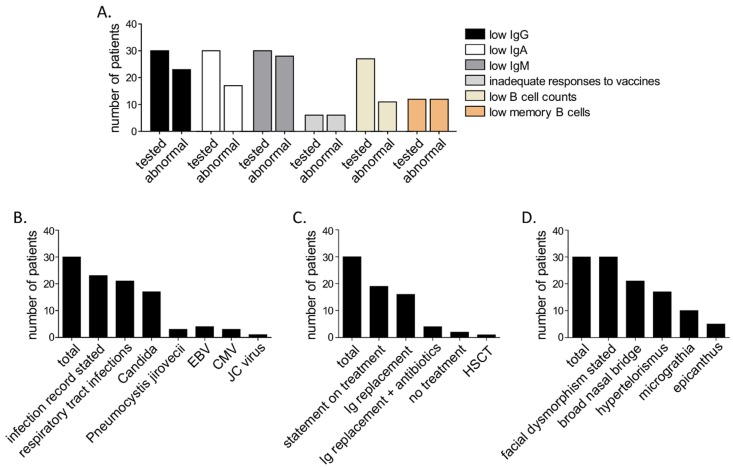
Summary of characteristics of all (*n* = 30, including the here presented patients), so far reported patients with ICF2. (**A**) Summary of immunological findings, (**B**) Infections, (**C**) treatment and (**D**) facial dysmorphism.

**Table 1 diseases-07-00034-t001:** Characteristics of three patients with ICF2.

Patient	Sex	Year of Birth	Age at First Diagnosis of PID/First Immunological Evaluation	Facial Anomalies	Motor Development	Intellectual Development	Infections	Other
P1	M	1992	3	broad flat nasal bridge, micrognathia	normal	Attention deficit hyperactivity disorder, treatment with methylphenidate till the 15th year of life, otherwise normal, work without difficulty	sepsis after birth, three pneumonias (*H. influenzae*), recurrent upper respiratory tract infections (bronchitis, otitis media, sinusitis)	atopic dermatitis
P2	F	1996	7	broad flat nasal bridge	normal	Selective mutism in early childhood, otherwise normal, work without difficulty	Atypical mycobacteriosis, recurrent shingles, prolonged fever after vaccination with MMR	atopic dermatitis, scoliosis, nail clubbing, focal bronchial malformation with absence of bronchial glands and cartilage
P3	F	1996	7	broad flat nasal bridge	normal	Selective mutism in early childhood, otherwise normal, work without difficulty	none	atopic dermatitis, idiopathic epileptic seizures

**Table 2 diseases-07-00034-t002:** Immunological findings in three patients with immunodeficiency, centromeric instability and facial anomalies syndrome 2 (ICF2).

Immunological Test	P1	P2	P3	Normal Range (or Control Value)
Year of birth	1992	1996	1996	
Serum Ig concentration/year of test	1997	2002	2002	
-IgG (g/L)	2.9	10.1	9.67	5.5–12.3
-IgA (g/L)	0.73	1.81	1.16	0.23–1.8
-IgM (g/L)	0.11	0.16	0.13	0.37–1.6
-IgE (IE/mL)	8.00	2	2	3–75
-IgG1 (g/L)	2.00	8.56	8.14	4.2–9.9
-IgG2 (g/L)	0.58	1.05	1.09	0.63–3.5
-IgG3 (g/L)	0.17	0.74	0.69	0.17–0.88
-IgG4 (g/L)	0.15	0.02	0.02	0.01–1.2
Lymphocyte subset analysis/year of test	2017	2012	2012	
Absolute lymphocyte count (cells/µL)	2193	1755	1710	1100–4500
Lymphocytes (% leukocytes)	43	27	30	20–44
CD19 B cells (% lymphocytes)	0,9	14.6	14.9	4.3–23.1
CD27-IgM+IgD+ naive B cells (% lymphocytes)	NM	13.8	14.1	2.6–15.7
CD27+IgM+IgD+ IgM Mem/Marg. Zone B cells (% lymphocytes)	NM	0.4	0.3	0.2–12.3
CD27+IgM-IgD-switched mem. Post GC B cells (% lymphocytes)	NM	0.3	0.3	1.9–30.4
CD38++IgM++ transitional B cells (% lymphocytes)	NM	0.3	1.1	0.6–3.5
CD38+++IgM+/− plasma cells (% lymphocytes)	NM	0.2	0.3	0.4–3.6
CD21lowCD38low B cells (% lymphocytes)	NM	0.1	0.1	4–26
CD16+CD56+CD3- NK cells (% lymphocytes)	1.4	4.7	5.9	7–31
CD3+ T cells (% lymphocytes)	92.4	68.8	66.6	55–83
CD3+CD4+ T helper cells (% lymphocytes)	7.1	55.9	55.6	27–53
CD3+CD8+ cytotoxic T cells (% lymphocytes)	84.5	11.5	9	19–34
CD4+CD45RO+ CD4 memory cells (% lymphocytes)	5.7	17	19.8	11–44
CD4+CD45RA+ naive CD4+ T cells (% lymphocytes)	1.9	40.1	37.9	21–75
CD4+CD45RA+CD31+ thymus emigrated CD4+ T cells (% lymphocytes)	1.7	36.5	35	19.4–60.9
CD8+CD27-CD28- late CD8+ effector cells (% lymphocytes)	5.4	1.3	0.3	2.9–16
CD8+CD27+CD28- CD8+ effector cells (% lymphocytes)	20	1.8	1.2	2.6–58
Lymphocyte proliferation assay				
-Medium control (% compared to control)	126	102	100	100
-Phytohemagglutinin (PHA) (% compared to control)	71	134	136	100
-Concanavalin A (ConA) (% compared to control)	80	128	113	100
-Pokeweed *mitogen* (PWM) (% compared to control)	80	141	151	100
-Tuberculin purified protein derivative (PPD) (% compared to control)	120	122	110	100
-Interleukin 2 (IL-2) (% compared to control)	84	129	130	100
-Anti-CD3-antibody (% compared to control)	118	148	154	100
NK cell activity				
-Natural cytotoxicity (1:30) (% compared to control)	29	20	47	100
-ADCC (1:30) (% compared to control)	26	67	67	100

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
