# Peer review of "Progressive Immunodeficiency with Gradual Depletion of B and CD4+ T Cells in Immunodeficiency, Centromeric Instability and Facial Anomalies Syndrome 2 (ICF2)"

_diseases, 2019, doi:10.3390/diseases7020034_

Round 1
Reviewer 1 Report
In this manuscript Sogkas et al report on 3 siblings with a
homozygous mutation in ZBTB24. Full clinical history is given, including immune
system composition/function over time. These findings are accompanied with a
review of available immunological data of all other ICF2 patients described to date,
and discussed. Of interest, as has been described for ICF1 patients but not ICF2
patients, a gradual CD4 T cell lymphopenia was observed in one of the patients.
Overall these data will be of interest to those studying ICF and clinicians
with ICF patients.
Minor comments:
The authors could do a better job in the text in referring to where the data is provided to support their statements concerning their patients (much of which is in the supplemental data)
I was wondering about the total blood IgG levels in the
supplemental exel file. Are these patients not on IVIG? If so an asterisk to point
out that it is not physiological would be useful.
Author Response
Only patient P1 receives immunoglobulin replacement treatment as indicated in text and Table S1, where all known patients with ICF2 are listed. We have added a `*` in supplement file 1, indicating serum-immunoglobulin measurements under immunoglobulin-substitution treatment.
Reviewer 2 Report
Sogkas G and others describe 3 siblings with ICF2 due to homozygous ZBTB24 gene mutation but with different clinical presentations, implying the epigenetic modification effect. Long-term changes are provided in this family. The introduction section is well organized. The cases are described clearly. The literature review is good and comprehensive. Although the mutation found in this family has been described previously, and there is no clear genotype-phenotype correlation in ICF2, the information provides here indeed provide more details about ICF2 patients. More research will be benefit from such accumulated information.
Author Response
We think reviewer 2 is satisfied with the manuscript.
